# Evolution of the Pattern of Spatial Expansion of Urban Land Use in the Poyang Lake Ecological Economic Zone

**DOI:** 10.3390/ijerph16010117

**Published:** 2019-01-04

**Authors:** Yang Zhong, Aiwen Lin, Zhigao Zhou

**Affiliations:** 1School of Resources and Environmental Sciences, Wuhan University, Wuhan 430079, China; zhongyang9093@163.com (Y.Z.); leehong@whu.edu.cn (Z.Z.); 2Key Laboratory of Geographic Information System, Wuhan University, Wuhan 430079, China

**Keywords:** Poyang Lake eco-economic zone, nighttime lighting data, comparative method based on ancillary data, Spatial expansion pattern, urban land

## Abstract

To grasp the evolutionary characteristics and regularity of urban land expansion patterns in the Poyang Lake Ecological Economic Zone, this study, based on nighttime lighting data, uses the Landsat series satellite simultaneous data and cluster analysis to correct the Defense Meteorological Satellite Program–Operational Linescan System (DMSP-OLS) nighttime lighting data and then uses the auxiliary data-based comparison method to determine the threshold for extracting the urban built-up area. Based on this threshold, a total of eight typical landscape pattern indicators, including landscape total area, total patches number, patches density, maximum patches index, and agglomeration index, etc., are selected. Next, the landscape spatial pattern analysis method and standard deviation ellipse method are used. The results show the following: (1) In 1992–2013, urbanization in the Poyang Lake Ecological Economic Zone expanded rapidly. The urban built-up area increased by 8.13 times, the number of plaques increased by 1.5 times, and the shape complexity of landscape plaques gradually increased. There is a large correlation between the changes in the total boundary length, and the average boundary density, the average annual growth rate of the two is 21.33% and 17.45%. (2) The two indicators of maximum plaque index and aggregation index show a downward trend year by year. However, there are some fluctuations and irregularities in the evolution of the total landscape area, total plaque number and plaque density. (3) The long axis and the short axis of the standard deviation ellipse of the Poyang Lake Ecological Economic Zone show small variation during the inspection period and generally have an elliptical shape. The movement of the center of gravity is mainly from the southwest to the northeast, but the migration of the center of gravity is relatively small. Based on this, this paper proposes three countermeasures and suggestions as a guide to promote the optimization and development of the spatial expansion pattern of the Poyang Lake eco-economic zone.

## 1. Introduction

At present, the development and construction of urban agglomerations in China are proceeding rapidly. Urban agglomeration refers to a network with more than one megacity as the core and a base of at least three large cities, which rely on networks of transportation and communication infrastructure within a specific geographical area. The area is compact and economically connected and eventually forms a highly urbanized and highly integrated urban group [1]. As an inevitable outcome of such development of new industrialization and urbanization, urban agglomeration has become a new regional unit for the country to participate in global competition and international division of labor. It is also the main battlefield for the construction of the “Belt and Road”. The development of urban agglomerations is a new type of town within the country. It has a dominant position in the development of economic and social development [2]. Under the background of economic globalization and regional integration, urban agglomerations have become an important form of spatial organization for the distribution of international productivity and the division of labor. Their evolution is increasingly related to the overall progress of a region or a country, and even the world economy, society and culture [3].

Currently, the study of the spatial pattern evolution of urban land expansion in urban agglomerations is one of the hot spots in academia. Research areas have mainly been in China [4,5,6,7] and South Asia [8,9]. In terms of research methods, the index method [10], the landscape pattern analysis method [11,12], geographically weighted regression [13], gravity modelling [14,15], and the social network analysis method [16] are mainly used. The Poyang Lake Ecological Economic Zone is a special economic zone that hosts Poyang Lake in Jiangxi Province as the core and the Poyang Lake City Circle as the core strategic concept for protecting the ecology and developing the economy. It has a special internal structure and evolutionary process of its spatial pattern. Current academic research on the Poyang Lake Ecological Economic Zone has focused on soil composition analysis [17,18], land use [19,20,21,22], economic linkages [23,24], and urbanization levels [25,26].

DMSP-OLS nighttime light data were obtained at night by the OLS (Operational Linescan System) sensor of the DMSP (Defense Meteorological Satellite Program) satellite used to record worldwide nighttime light intensity [27]. Because nighttime light (NTL) intuitively reflects the intensity of human activity, it has great potential for studies in global cities, and global changes, etc. [28]. In addition, since the data also host strong spatial and intensity information through time, it is also considered suitable for monitoring and identifying urban land types that have different extended intensity change trajectories on a large scale. At present, domestic and foreign scholars have achieved significant research results based on DMSP/OLS night lighting data. These research results mainly include urbanization expansion [29,30,31,32,33], population and economic estimations [34,35,36], power consumption estimations [37,38,39], and the spatial distribution characteristics of carbon emissions [40,41,42]. In summary, academic research based on DMSP/OLS nighttime lighting data has been applied less to the spatial expansion pattern of the Poyang Lake Ecological Economic Zone.

Therefore, based on the above research status, this paper uses DMSP/OLS nighttime lighting data supported by Landsat-TM data and the 1 km Chinese land use/coverage grid dataset. The research method is to study the evolution of the spatial pattern of the urban land expansion in the Poyang Lake ecological economic zone from 1992–2013. The purpose is to discover the spatial pattern evolution characteristics of urban land expansion in the Poyang Lake Ecological Economic Zone, which is statistically rigorous. This study can not only enrich the depth and breadth of relevant research, but also provide some guidance for the optimization and development of the problems in the urban expansion process of Poyang Lake Ecological Economic Zone. Consequently, it has practical value.

## 2. Research Area and Data Source

### 2.1. Research Area

The Poyang Lake Ecological Economic Zone is a special economic zone that hosts Poyang Lake in Jiangxi Province as its core, and the Poyang Lake City Circle is an important strategic concept for protecting the regional ecology and developing the economy. The study area includes Nanchang, Jingdezhen, Yingtan, and Jiujiang, Xinyu, Fuzhou, Yichun, Shangrao, Ji’an City, a total of 38 counties (cities, districts), and the land area is 51,200 square kilometers, the specific administrative scope and location are shown in Figure 1. The region, which accounts for 30% of Jiangxi’s land area, hosts nearly 50% of the province’s population and generates more than 60% of its total economic output. It is the region with the strongest comprehensive strength and the greatest development potential in Jiangxi. Located on the south bank of the middle and lower reaches of the Yangtze River and the northern part of the Jiangxi Province, Poyang Lake is the largest freshwater lake in China. It is the only lake of the four major freshwater lakes that is not eutrophic and is also an important wetland with world influence.

On 12 December 2009, the State Council of China officially approved the “Poyang Lake Ecological Economic Zone Plan”, marking the official rise of the Poyang Lake Ecological Economic Zone as a national strategy. This is also the first regional development plan of Jiangxi Province that has been included in the national strategy since the founding of the people’s Republic of China. It is a major milestone in the history of Jiangxi’s development and has great and far-reaching significance for realizing the new leap of Jiangxi’s rise. In addition, the Poyang Lake Ecological Economic Zone is also the direct hinterland of important economic sectors, such as the Yangtze River Delta, the Pearl River Delta, and the West Coast Economic Zone. It is an important growth area that is accelerating in the central region. Further, it is an important base for the central manufacturing industry and the three major innovation regions of China. It has good conditions for promoting the coordinated development of regional ecology and the economy. Accelerating the construction of the national Poyang Lake eco-economic zone as an important national strategy is conducive to exploring a new path of coordinated development of the ecology and economy, facilitating the exploration of a new model for the comprehensive development of the Great Lakes Basin, and facilitating the construction of a national strategy to promote the rise of the central region. The new fulcrum is conducive to establishing a new image of China’s adherence to the path of sustainable development.

### 2.2. Data Sources

#### 2.2.1. DMSP/OLS Night Light Data

This study used a 22-year (1992–2013) DMSP-OLS non-radiation-calibrated night-time stable lighting dataset downloaded from the National Geophysical Data Center. The data contain towns and other types of stable lights and have been strictly processed to remove the effects of fire, sunlight, and clouds. For the research period, four images (F101992, F141999, F162006, and F182013) were selected. The frequency distribution of gray values from year-to-year for the Poyang Lake eco-economic zone are shown in Figure 2. At the same time, to verify the accuracy of the model results, the nighttime lighting data were corrected using the Landsat series satellite synchronization data. The data were obtained from the International Scientific Data Service Platform of the Computer Network Information Center of the Chinese Academy of Sciences.

#### 2.2.2. Land Use Data

The data were derived from the land use/coverage grid dataset with a spatial resolution of 1 km in China for the years 1995, 2000, and 2005, published on the Earth System Science Data Sharing Platform. The land use/coverage classification system of the data includes six first-class types including “cultivated land”, “forest land”, and “urban and rural areas, industrial and mining, and residential land”, among which “urban and rural areas, industrial and mining, and residential land” includes three secondary types including “urban land”, “rural settlements” and “other construction sites” [25]. Therefore, the secondary “Urban Land” data for 1995, 2000, and 2005 were obtained from the dataset and were then used as auxiliary data to extract the optimal threshold of the urban built-up area of the Poyang Lake Ecological Economic Zone from 1992 to 2013.

#### 2.2.3. Statistical Data

Statistical data, such as the built-up area of the city, were derived from the “China Urban Statistical Yearbook” for each corresponding year. The other basic data are from the “Jiangxi Statistical Yearbook” for each corresponding year, as well as statistical yearbooks from other counties and cities within the study area.

## 3. Research Method

### 3.1. Cluster Analysis of Raw Lighting Data

To ensure the determination of the threshold for extracting the urban built-up area is more accurate, cluster analysis of DMSP/OLS nighttime lighting data in the three years 1995, 2000, and 2005 was carried out to grasp the distribution characteristics of nighttime lighting data brightness values. The statistical results of the cluster analysis are shown in Table 1.

### 3.2. Extraction of the Urban Built-Up Area

Based on a comprehensive comparison of long-term urban built-up area extraction methods based on DMSP/OLS nighttime lighting data in China and abroad, the threshold method was selected for extracting urban built-up areas. Considering that the threshold of the urban built-up area is the key to the threshold method, the auxiliary data-based comparison method included in the threshold extraction method can better meet the needs for understanding the long-term sequence of the spatial pattern evolution for the Poyang Lake Ecological Economic Zone. Therefore, this paper uses the comparison method based on auxiliary data to select the optimal threshold.

### 3.3. Use Spatial Pattern Analysis

This paper uses the landscape index method from landscape ecology [26] to analyze the urban built-up area of the Poyang Lake Ecological Economic Zone for the years 1992, 1999, 2006, and 2013. This study accounts for the complex and diverse landscape types of the urban agglomeration data for the Poyang Lake Ecological Economic Zone extracted from the DMSP-OLS lighting data. Parameters include the total landscape area (TA), total number of patches (NP), and patches density (PDh), Largest Patch Index (LPI), total boundary length (TE) and average boundary density (ED), etc. There are a total of eight landscape pattern indicators; the meaning of each is shown in Table 2. The landscape pattern analysis software FRAG-STATS 4.2 was used to calculate the eight indicators.

### 3.4. Standard Deviation Ellipse Method

The Standard Deviational Ellipse (SDE) method is a spatial statistical method that reveals the multi-faceted features of variable spatial distribution [43]. The center of gravity of the SDE represents the relative position of the spatial distribution of economic elements. The azimuth of the SDE reflects the main trend direction of the distribution. The long axis and short axis lengths of the SDE are standard distance, which respectively indicates the degree of dispersion of economic elements in the main trend direction and the degree of dispersion of economic elements in the secondary direction. The standard deviation ellipse can reveal the overall characteristics of the spatial distribution of geographic elements, mainly reflecting the center of gravity, main trend, and secondary trend of the elements in the two-dimensional spatial distribution through the center, along the main axis, along the auxiliary axis, and the azimuth angle. These include discreteness and main trend direction [44]. The center of the ellipse is the average center of the spatial distribution, which is used to reflect the relative position of the center of gravity of the geographic feature layout and its changes. The long and short axes of the standard deviation in the X and Y directions represent the main trend direction of the spatial element layout and the degree of dispersion of the secondary direction. The azimuth reflects the main trend direction of its distribution. The area of the ellipse represents the concentration or divergence of the spatial distribution of geographic features.

## 4. Results and Discussion

As shown in Table 3, the calculated results from the FRAG-STATS 4.2 software are statistically compiled, and we can also clearly find the changing characteristics of various landscape pattern indicators in 1992–2013.

### 4.1. Evolution of the Spatial Pattern of Urban Land Use in the Poyang Lake Ecological Economic Zone

#### 4.1.1. Changes in the Total Landscape Area (TA), Total Plaque Number (NP), and Plaque Density (PDh)

The total landscape area (TA), total patches number (NP), and patches density (PDh) all show yearly growth. This indicates that the urban land area and the number of towns in the Poyang Lake Ecological Economic Zone increased significantly. The number of towns and the area of urban land have been significantly increased. Specifically, the average annual growth rate of these three indicators reached 36.98%, 6.82%, and 6.82%, respectively. At the same time, it can be clearly found that the increase in 2003–2013 is higher than that in 1992–2002. In particular, the increase in 2010–2013 is the most obvious. This is because the China State Council officially approved the “Poyang Lake Ecological Economic Zone Plan” on 12 December 2009. Therefore, the development of the Poyang Lake Ecological Economic Zone has received various funds and policy support from the national level, and its urbanization speed has also been significantly improved. Specifically, the growth of isolated urban sectors is more obvious, indicating that during the 22 years from 1992 to 2013, a large number of emerging towns appeared in the Poyang Lake Ecological Economic Zone. However, from 1992–1998 and 2003–2010, the fluctuations of the two stages are the most obvious. At the same time, the fluctuations of these three indicators also show obvious correlations, which further reflects the development of the Poyang Lake Ecological Economic Zone. With the development and growth of instability and volatility, it is urgent to adjust according to scientific planning.

#### 4.1.2. Changes in the Total Index Length (TE), Average Boundary Density (ED), and Landscape Shape Index (LSI)

The total boundary length (TE), average boundary density (ED), and landscape shape index (LSI) all showed sustained growth characteristics. The average annual growth rates of total boundary length (TE) and average boundary density (ED) are 21.33% and 17.45%, respectively. The growth of these two indicators from 2003–2013 is significantly higher than the increase from 1992–2002. The average annual growth rate of the landscape shape index (LSI) is 2.01%. The shape complexity of the Poyang Lake Ecological Economic Zone gradually increased, and the indicator shows that the concentration of the Poyang Lake Ecological Economic Zone also increased, and, therefore, their mutual influence increased. This is because the towns in the Poyang Lake Ecological Economic Zone are basically distributed around Poyang Lake. Historically, these towns have been close to each other for a long time. In addition, with the obvious improvement of traffic accessibility in the Poyang Lake Ecological Economic Zone, the internal relations between towns are converging. The fluctuations of development and change are also obvious, especially from 1992–1998 and 2003–2010, which fluctuate greatly. This reflects the fact that in the development of the Poyang Lake Ecological Economic Zone, the driving role of the core cities is not prominent.

#### 4.1.3. Evolution of the Largest Patch Index (LPI) and Aggregation Index (AI)

The largest plaque index (LPI) and aggregation index (AI) generally show a decreasing yearly trend. The fluctuation of the maximum patches index (LPI) is more obvious, especially in 2001–2013. It shows irregular and disordered characteristics indicating that there is uncertainty in the change in the core urban area of the Poyang Lake Ecological Economic Zone. As the core city and the only big city in the Poyang Lake Ecological Economic Zone, Nanchang is the core of the development of the Poyang Lake Ecological Economic Zone. However, Nanchang City has a certain gap compared with the neighboring cities of Wuhan and Changsha in terms of city size and economic aggregate. Therefore, it is necessary to concentrate on promoting the development of Nanchang City, and finally to better play the core role of Nanchang City in the development of Poyang Lake Ecological Economic Zone. The change in the aggregation index (AI) shows a gradual yearly decline, which also reflects the increasingly close relationship between the urban areas of the Poyang Lake Ecological Economic Zone and the continuous improvement of the urbanization level.

### 4.2. Spatial Evolution of Urban Land Expansion of the Poyang Lake Ecological Economic Zone

This paper represents a simple model called the Standard Deviational Ellipse (SDE) model, which is embedded in the popular GIS software ArcGIS [45]. The standard deviation ellipse, as a tool for measuring spatial distribution, is mainly used to determine the range and directional factors of spatial distribution features. It is one of the most important tools for studying the distribution and evolution of spatial elements. Through using the standard deviation ellipse of nighttime lighting data of the Poyang Lake Ecological Economic Zone, the direction and concentration of the Poyang Lake Ecological Economic Zone in space could be further clarified. Therefore, to further analyze the spatial direction of urban land expansion in the Poyang Lake Ecological Economic Zone, a standard deviation ellipse was drawn for the area of urban expansion in the Poyang Lake Ecological Economic Zone. Through changes in the area of the ellipse and the movement of the center of gravity, the direction of the expansion of urban land use was clarified.

Figure 3 and Figure 4 show that the long axis and short axis of the standard deviation ellipse of the Poyang Lake Ecological Economic Zone change little over the inspection period. The general form is an elliptical shape, and the spatial distribution generally shows a “southwest-northeast” orientation. The standard deviation of the urban system scale is slightly stronger than the short axis or the long axis of the ellipse. This indicates that urban growth in the east-west direction is slightly stronger than in the north-south direction. This is because the urban distribution of the Poyang Lake Ecological Economic Zone is mainly concentrated in the east and the west. For example, Nanchang City, the core city of the Poyang Lake Ecological Economic Zone, is located in the western part of the region. Therefore, the urban growth in the east-west direction is stronger than the north-south direction. From 1999–2013, the variation of the ellipses was larger than the variation of the standard deviation ellipse from 1992–1999. It is apparent that after the 21st Century, the economic expansion of the Poyang Lake Ecological Economic Zone increased significantly. In addition, this also reflects that the implementation of the “Poyang Lake Ecological Economic Zone Plan” has, indeed, accelerated the development of the Poyang Lake Ecological Economic Zone.

In terms of a gravity shift during the study period, the center of gravity moved mainly from southwest to northeast. The coordinates of the urban expansion center were transferred from (116.09 E, 28.47 N) in 1992 to (116.10 E, 28.47 N) in 2013. The magnitude of change is small. Specifically, the focus of urban expansion of the Poyang Lake Ecological Economic Zone is located in Nanchang County. This is because Nanchang County is a county under the jurisdiction of Nanchang City. It is closely connected with Nanchang City. Nanchang City is the core city of Poyang Lake Ecological Economic Zone. In addition, there are also prefecture-level cities, such as Fuzhou and Yingtan around Nanchang County. Therefore, the area around Nanchang County is the concentrated area of urban distribution in the Poyang Lake Ecological Economic Zone. From 1992–2013, the center of gravity shifted 0.5 km, and the average annual transfer to the northeast was 0.02 km. From 1992–1999, the transfer speed was very slow, showing almost no transfer. In contrast, the transfer speed from 1999–2006 accelerated significantly to 0.1 km per year. The northeast transfer speed in 2006–2013 was also relatively slow, shifting to the southeast at an annual rate of 0.4 kilometers. Throughout the study period, the focus of the Poyang Lake Ecological Economic Zone gradually moved to the northeast. The urban areas at the fringe of the urban system in the central and western regions grew faster than other regions. The northern part of Nanchang County is the region with the most significant growth in urban scale.

### 4.3. Evolution of Urban Spatial Patterns in the Poyang Lake Ecological Economic Zone

In this study, DMSP-OLS nighttime stable lighting data were used to extract urban land use in the Poyang Lake Ecological Economic Zone for the years 1992, 1999, 2006, and 2013. This was also done for Nanchang City using the comparative method based on ancillary data, as shown in Figure 5, during the period 1992 to 2013, with the increase of the year, the urban built-up area of the entire Poyang Lake Ecological Economic Zone has been significantly increased. Specifically, the urban expansion of the Poyang Lake Ecological Economic Zone is getting faster and faster. For example, the urban expansion rate in 1999 to 2006 was faster than that in 1992 to 1999, and the urban expansion rate in 2006 to 2013 was faster than 1999 to 2006. In Figure 5, the expansion rate of urban land use in the core city of Nanchang is the most obvious. At the same time, considering that Nanchang City is the core city of Poyang Lake Ecological Economic Zone, it also draws the evolution map of the urban built-up area of Nanchang City in 1992, 1999, 2006, and 2013. Specifically, the urban land area of Nanchang City increased from 65 km^2^ in 1992 to 249.5 km^2^ in 2013, which showed an increase of 3.84 times over 22 years. The strength increased greatly, which has prompted Nanchang City to play a leading role as a core growth pole in the development of the Poyang Lake Ecological Economic Zone.

## 5. Conclusions

The research results of this paper show the following. (1) Between 1992 and 2013, the urbanization of the Poyang Lake Ecological Economic Zone expanded rapidly, the urban built-up area increased by 8.13 times, the number of patches increased by 1.5 times, and the patch density also increased greatly. (2) The complexity of the patches’ shape gradually increased, as there were more emerging small towns. The average annual growth rates of the total boundary length and average boundary density were 21.33% and 17.45%, respectively. (3) The two indicators of maximum patches index and aggregation index were conclusive. They show a yearly decline. However, indicators including total landscape area, total patches number, and patches density show a degree of fluctuation and disorder. (4) The long axis and short axis of the standard deviation ellipse of the Poyang Lake Ecological Economic Zone changed little over the inspection period. In terms of a gravity shift during the study period, the center of gravity moved mainly from southwest to northeast.

Based on this, this paper proposes three countermeasures and suggestions to promote the optimization of development of the Poyang Lake Ecological Economic Zone: (1) Implement existing plans, actively integrate these plans into the integrated development of urban agglomerations in the middle reaches of the Yangtze River, and formulate other relevant plans with high specifications. Implementing this measure ensures that the planning of the Poyang Lake eco-economic zone has been implemented, and finally the development and construction of the Poyang Lake Ecological Economic Zone have been guided by scientific planning. (2) Concentrate on promoting the development of Nanchang City, enhancing the competitiveness of Nanchang City, and enhancing the core role and radiation of Nanchang City. By implementing this countermeasure, we can accelerate the development of Nanchang City, the core city of Poyang Lake Ecological Economic Zone, and finally ensure that Nanchang City can better play its core role. (3) We should not only pay attention to the development and rise of large and medium-sized cities but also the need to rationally improve the development speed of small towns and eventually build a proper and orderly urban structure. The implementation of the countermeasures can ensure that the Poyang Lake Ecological Economic Zone forms a scientific and rational urban structure, and it ultimately guarantees the scientific and orderly development of the cities in the region.

Poyang Lake is the only member of the World Life Lake Network in China. It is an internationally important wetland and the largest wintering migratory bird habitat in Asia. It has received extensive international attention. Therefore, to build the Poyang Lake Ecological Economic Zone, it is necessary to establish the principle of ecological environmental protection first, focus on protecting and rehabilitating the lake ecosystem, rationally develop the area around the lake, strictly implement the scientific plan, and finally ensure that the towns around the Poyang Lake eco-economic zone have coordinated development. The comprehensive development and governance of the Great Lakes Basin is a major issue in the world today. The research work of this paper can provide some reference for the content that needs to be analyzed in the comprehensive development and governance of the Great Lakes Basin, that is, the characteristics of urban land expansion and evolution in the region.

Even though DMSP/OLS night lighting data has obvious advantages in the field of urbanization research, the relatively low resolution of DMSP/OLS nighttime lighting data leads to underrepresentation of county towns and small towns. The accuracy of the data in smaller urban areas is not high, and more detailed quantitative research through the use of higher resolution data can improve this situation. This must be expanded in subsequent research.

## Figures and Tables

**Figure 1 ijerph-16-00117-f001:**
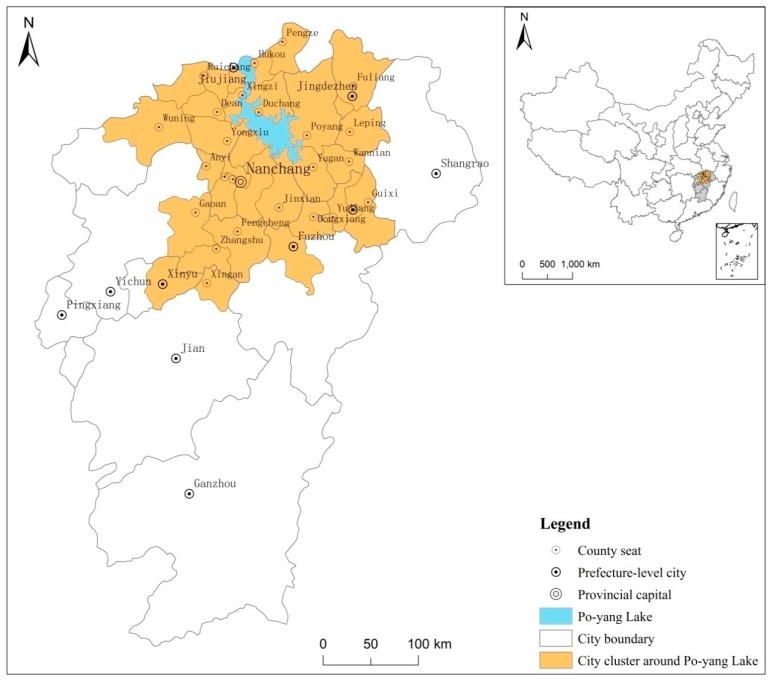
Poyang Lake Ecological Economic Zone Location Map.

**Figure 2 ijerph-16-00117-f002:**
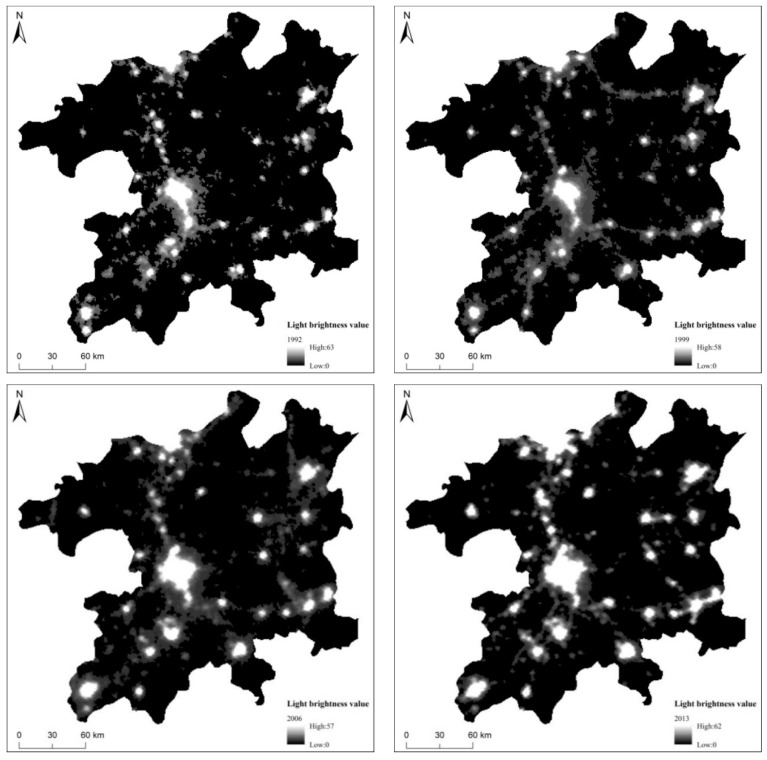
Poyang Lake Ecological Economic Zone nighttime light data from 1992, 1999, 2006, and 2013.

**Figure 3 ijerph-16-00117-f003:**
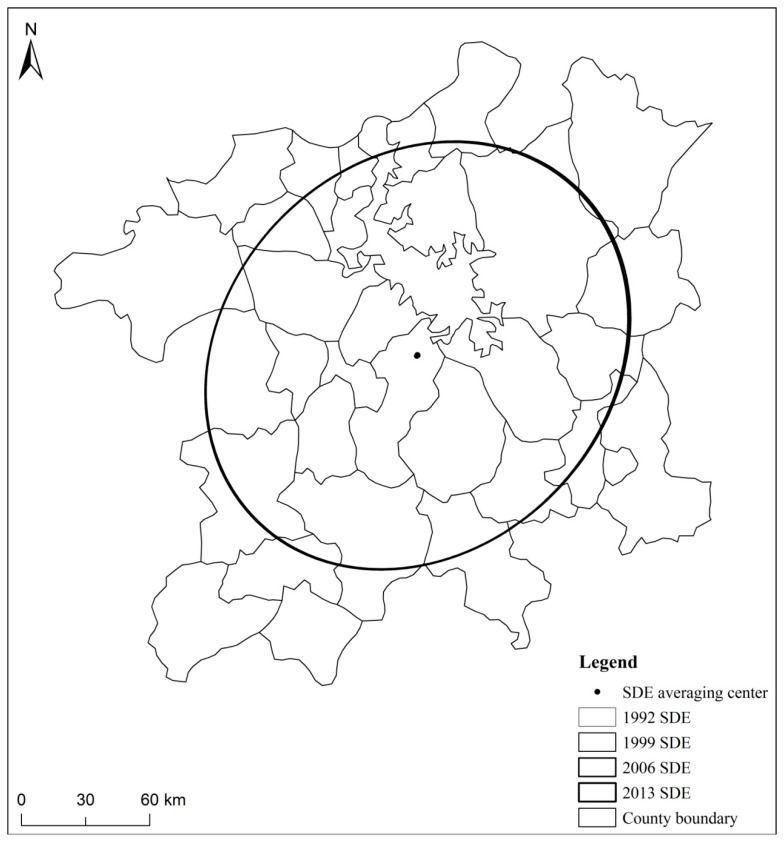
Standard deviation ellipse of urban land expansion in the Poyang Lake Ecological Economic Zone.

**Figure 4 ijerph-16-00117-f004:**
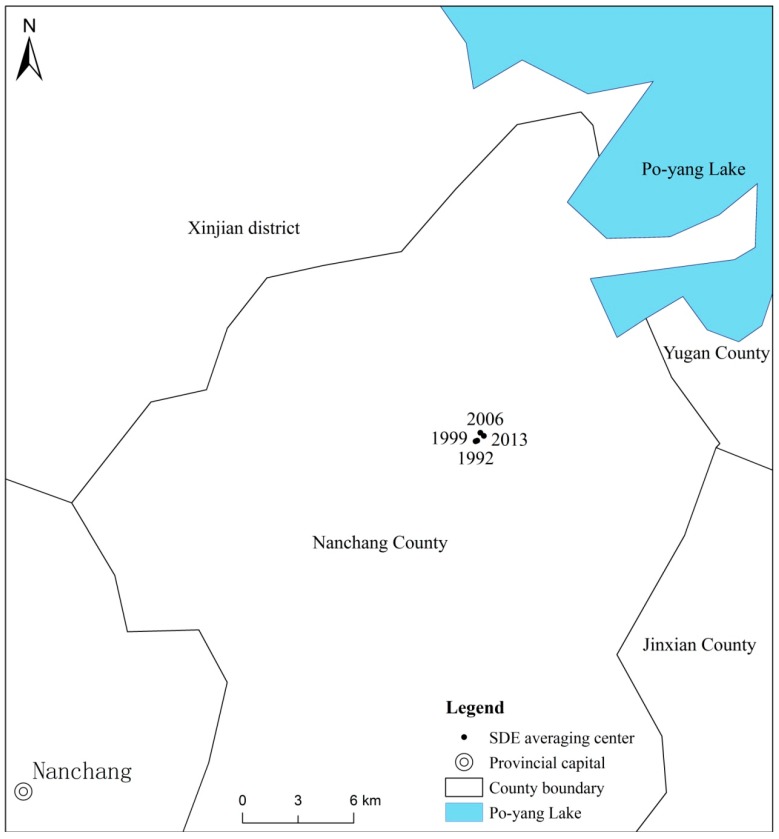
Standard deviation elliptical average center of four years of urban land expansion in the Poyang Lake Ecological Economic Zone.

**Figure 5 ijerph-16-00117-f005:**
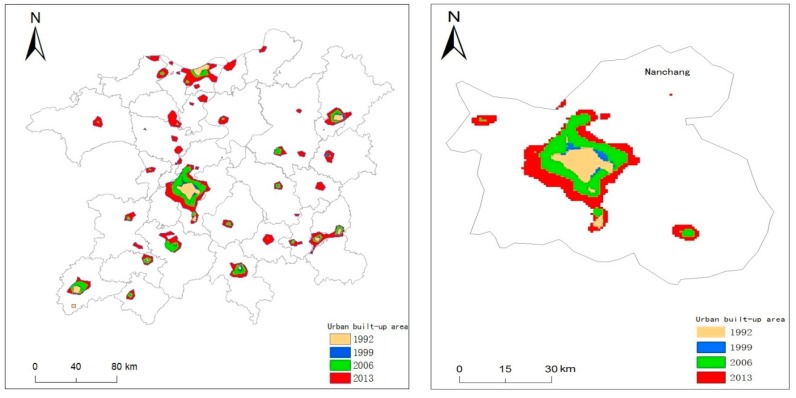
Comparison of the urban built-up area between the Poyang Lake Ecological Economic Zone and Nanchang City in 1992, 1999, 2006 and 2013.

**Table 1 ijerph-16-00117-t001:** Nighttime light data clustering analysis statistics for 1995, 2000 and 2005.

Years	Cluster	Brightness Value (Average)	Proportion	Total Size
1995	1	6.87	78.2%	2968
	2	26.43	21.8%	829
2000	1	30.83	17.5%	620
	2	8.13	82.5%	2928
2005	1	8.81	77.5%	3870
	2	33.73	22.5%	1123

**Table 2 ijerph-16-00117-t002:** Landscape pattern index.

Landscape Index	Shorthand	Description
Total Area	TA	The sum of the areas of all patches
Number of Patches	NP	The total number of all patches in the landscape
Patch Density per 100 km^2^	PDh	Number of patches in an area of 100 km^2^
Largest Patch Index	LPI	The largest patches in a patches type as a percentage of the total landscape area
Total Edge	TE	Total patches length of all plaques
Edge Density	ED	Length of patches boundary per unit area
Landscape Shape Index	LSI	Patches landscape shape indicator
Aggregation index	AI	The number of similar adjacencies of the corresponding type divided by the maximum value when the type is most confluent as a patch (multiplied by 100 to produce a percentage)

**Table 3 ijerph-16-00117-t003:** Calculated results of the landscape pattern of the Poyang Lake Ecological Economic Zone from 1992 to 2013.

Year	TA (km^2^)	NP	PDh	LPI (%)	TE (km)	ED (m/km^2^)	LSI	AI
1992	400.9	18	4.13	99.21	27.60	6.33	3.51	98.93
1993	297.83	14	3.21	99.41	19.44	4.46	3.41	99.01
1994	546.05	19	4.36	98.93	32.96	7.57	3.58	98.89
1995	485.89	20	4.59	99.05	30.48	7.00	3.55	98.91
1996	420.45	16	3.67	99.17	24.72	5.67	3.48	98.96
1997	318.92	14	3.21	99.37	20.80	4.77	3.43	99.00
1998	370.02	14	3.21	99.27	22.48	5.15	3.45	98.98
1999	415.18	15	3.44	99.18	25.28	5.80	3.48	98.96
2000	522.7	18	4.13	98.97	30.16	6.92	3.54	98.92
2001	549.07	20	4.60	98.92	30.48	7.00	3.55	98.91
2002	959.69	29	6.66	98.12	54.00	12.39	3.83	98.71
2003	1085.34	27	6.20	97.88	57.04	13.09	3.86	98.68
2004	1562.93	37	8.49	96.94	80.24	18.42	4.14	98.48
2005	1083.03	26	5.97	97.88	53.36	12.25	3.82	98.72
2006	1338.01	29	6.66	97.38	62.48	14.34	3.93	98.64
2007	1856.26	41	9.41	96.37	89.60	20.57	4.25	98.40
2008	1879.55	40	9.18	96.32	90.72	20.82	4.27	98.38
2009	1465.91	34	7.81	97.13	72.32	16.60	4.05	98.55
2010	3066.42	44	10.1	93.97	133.52	30.65	4.78	98.01
2011	2845.29	39	8.95	94.44	125.12	28.72	4.68	98.08
2012	3058.89	42	9.64	94.02	133.44	30.63	4.78	98.00
2013	3662.85	45	10.33	92.84	157.12	36.06	5.06	97.79

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
