# Peer review of "Evolution of the Pattern of Spatial Expansion of Urban Land Use in the Poyang Lake Ecological Economic Zone"

_ijerph, 2019, doi:10.3390/ijerph16010117_

Reviewer 1 Report

In this paper the authors study the urban land expansion patterns of the Poyang Lake Ecological Economic Zone in China. The use of nighttime lighting data and analytical approach are commendable. However, I found this manuscript rather “descriptive” in nature regarding what happened over time (1992-2013), and it lacks some background information as to why such observed changes happened. Because of this, I would suggest the following for the authors to consider:

1. Expand the discussion under each of the results sub-sections to describe, based on the authors’ understanding of the study area, what might be the likely factors that fueled such changes.

2. The three suggested “countermeasures and suggestions” in the Conclusions section are too general. It would be helpful that the authors expand each of the three suggestions by describing what can be achieved if such actions are implemented.

A minor note is about the use of the term “plaque” and it seems to me that the authors use it and “patch” interchangeably (e.g., Table 2). It should be used consistently throughout the manuscript unless their meaning is different.

Author Response

Response to Reviewer 1 Comments

Point 1:  Expand the discussion under each of the results sub-sections to describe, based on the authors’ understanding of the study area, what might be the likely factors that fueled such changes.

Response 1: The extensions and supplements to Sections 4.1, 4.2, and 4.3 of the research results of the paper have been extended based on the opinions of reviewers.

Point 2: The three suggested “countermeasures and suggestions” in the Conclusions section are too general. It would be helpful that the authors expand each of the three suggestions by describing what can be achieved if such actions are implemented.

Response 2: The implementation effect of each countermeasure proposed in the conclusion section has been described in accordance with the recommendations of the reviewer.

Point 3: A minor note is about the use of the term “plaque” and it seems to me that the authors use it and “patch” interchangeably (e.g., Table 2). It should be used consistently throughout the manuscript unless their meaning is different.

Response 3: On the basis of following the recommendations of the reviewers for reviewing the manuscript, the term “patch” was consistently used in the content of the full-text paper, and the inconsistent words were modified.

Reviewer 2 Report

The authors use night lighting data with land use and other statistical data to determine patterns of urban development over time within an ecologically and economically significant region in China. The manuscript is interesting and generally well written (with some editorial attention needed).

Although I am not an expert in the particular methods used, the authors do a nice job in relaying the methodology and results.

Suggestions for improvement:

Be sure to explain terms that may be unfamiliar to readers who have little background in Chinese policy (e.g., "Belt and Road")

Make the significance/importance of this research even clearer

Discuss the implications of the findings even more. Even though ecological issues are mentioned throughout the manuscript, the authors do not really discuss how these urban growth patterns may impact the natural environment of the area.

Author Response

Response to Reviewer 2 Comments

Point 1: Be sure to explain terms that may be unfamiliar to readers who have little background in Chinese policy (e.g., "Belt and Road")

Response 1: Further explain the policy background and implementation significance of the Poyang Lake eco-economic zone planning included in this part of the study area.

Point 2: Make the significance/importance of this research even clearer.

Response 2: Appropriate additions and deletions have been made to the introduction of the paper in accordance with the recommendations of the reviewers, thus ensuring that the research significance and importance of this paper are clearer.

Point 3: Discuss the implications of the findings even more. Even though ecological issues are mentioned throughout the manuscript, the authors do not really discuss how these urban growth patterns may impact the natural environment of the area.

Response 3: The results and discussion of the paper have been further expanded and discussed based on the recommendations of the reviewers. This paper takes the Poyang Lake Ecological Economic Zone as the research object, and aims to study and discover the characteristics of the urban land expansion and evolution pattern in the Poyang Lake Ecological Economic Zone, and propose optimization strategies and suggestions.

Reviewer 3 Report

This is a good paper that needs some improvement.

First, define the term "plaques," and explain how it relates to FRAGSTATS.

Second, on page 8, you say that the Agglomeration Index declines. I understand this to mean greater sprawling development and lower density development. Yet, later the growth of Nanchang City is noted and urban area in the region has increased 8.13 times. Please explain this apparent contradiction.

Third, you should include population growth for Nanchang City and the region from 1992-2013.

Then ask, how has the percentage of population of Nanchang City compared to the region changed from 1992  to 2013?

Fourth, on page 11, the area of Nanchang City grew by 3.84 times, not 2.84 times.

Fifth, the conclusions need to be consistent. You want the core city of Nanchang to be stable.

Then why do you recommend growth for large and medium cities and even small towns?

Can the region create a network of cities without sprawling by relying on cars and trucks? A polynucleated region makes sense only if it is linked together by abundant mass transit.

The elliptical pattern is interesting, but you can lay the groundwork by noting that Von Thunen the first researcher on urban form, based his work on a circular city form. 

Is the elliptical pattern optimal? Or the circular pattern? Does it make a difference?

Finally, you should say something about how development in the region has located away from Poyang Lake. As China's largest freshwater lake, this is a very valuable body of water. 

Also, explain how this study is relevant and potentially helpful to other parts of China that are undergoing rapid urban development. 

Author Response

Response to Reviewer 3 Comments

Point 1: First, define the term "plaques," and explain how it relates to FRAGSTATS.

Response 1: Plaque is one of the basic concepts in landscape ecology, and it is also the basic unit of landscape type classification.  Fragstats is a powerful landscape pattern index calculation software that provides a variety of functional modules based on cell-based metrics, surface metrics, sampling strategies, functional metrics, metrics, etc. Therefore, plaque is the basic indicator for the calculation of various functional modules of the fragstats software.

Point 2: Second, on page 8, you say that the Agglomeration Index declines. I understand this to mean greater sprawling development and lower density development. Yet, later the growth of Nanchang City is noted and urban area in the region has increased 8.13 times. Please explain this apparent contradiction.

Response 2: The decline in the aggregation index reflects the acceleration of urbanization and the decline in urban density. The paper only mentions that the area of urban built-up areas in Nanchang has increased by 3.84 times instead of 8.13 times in 22 years in the “4.3 Evolution of urban spatial patterns in the Poyang Lake Ecological Economic Zone”. In addition, the paper mentioned in the “5. Conclusions” that the area of the entire Poyang Lake eco-economic zone increased by 8.13 times in 1992-2013, this is also positively related to the result of the decrease in the aggregation index.

Point 3: Third, you should include population growth for Nanchang City and the region from 1992-2013. Then ask, how has the percentage of population of Nanchang City compared to the region changed from 1992  to 2013?

Response 3: Dear reviewer, thank you very much for your review suggestion, "Third, you should include population growth for Nanchang City and the region from 1992-2013.  Then ask, how has the percentage of population of Nanchang City compared to the region changed from 1992 to 2013?” After detailed communication with my doctoral supervisor, we believe that the purpose of this paper is to discover the evolution of urban land use in the Poyang Lake eco-economic zone based on the perspective of night light, that is, the evolution of urban land expansion. The auxiliary data and research methods can well support the research results of this paper. At the same time, the discussion content of this research part is also more detailed and specific. Therefore, we think that these contents are closely related and sufficient to the title of this article.

Point 4: Fourth, on page 11, the area of Nanchang City grew by 3.84 times, not 2.84 times.

Response 4: The increase in the 22-year period of the urban built-up area of Nanchang City from 1992 to 2013 has been revised to 3.82 times.

Point 5: Fifth, the conclusions need to be consistent. You want the core city of Nanchang to be stable. Then why do you recommend growth for large and medium cities and even small towns? Can the region create a network of cities without sprawling by relying on cars and trucks? A polynucleated region makes sense only if it is linked together by abundant mass transit.

Response 5: The conclusions of the paper are consistent. The conclusions of the paper put forward three suggestions. The second proposal clearly emphasizes the need to concentrate on promoting the development of Nanchang City, so as to better play the core role of Nanchang City in the Poyang Lake Ecological Economic Zone. The third proposal proposes Actively develop large- and medium-sized cities, steadily expand the scale of development of small towns, and build a scientific and rational urban structure. Therefore, the second proposal does not conflict with the third proposal. But complement each other According to the current urban development status of the Poyang Lake Ecological Economic Zone, Nanchang City is the only large city in the region. The rest of the cities are small and medium-sized cities or towns. Therefore, there will be no spread in a certain period of time. In addition, the Poyang Lake Ecological Economic Zone is an urban agglomeration built around Poyang Lake. Its main towns are located in the area around Poyang Lake. It should also adopt the strategy of joint development of water transport, highway and air transport.

Point 6: The elliptical pattern is interesting, but you can lay the groundwork by noting that Von Thunen the first researcher on urban form, based his work on a circular city form. The second proposal clearly emphasizes the need to concentrate on promoting the development of Nanchang City, so as to better play the core role of Nanchang City in the Poyang Lake Ecological Economic Zone. Is the elliptical pattern optimal? Or the circular pattern? Does it make a difference?

Response 6: Thank you very much, I know that Von Thunen is the first researcher in the urban form, and I will also draw on his research and experience. The elliptical pattern is the best effect. The ellipse of this paper looks almost circular because the long and short axes of the standard deviation ellipse of the Poyang Lake eco-economic zone change little during the study period. Therefore, the standard deviation ellipses of the four years 1992, 1999, 2006 and 2013 appear to be approximately superimposed.

Point 7: Finally, you should say something about how development in the region has located away from Poyang Lake. As China's largest freshwater lake, this is a very valuable body of water. Also, explain how this study is relevant and potentially helpful to other parts of China that are undergoing rapid urban development.

Response 7: Poyang Lake is the only member of the World Life Lake Network in China. It is an internationally important wetland and the largest wintering migratory bird habitat in Asia. It has received extensive international attention. Therefore, to build the Poyang Lake Ecological Economic Zone, it is necessary to establish the principle of ecological environmental protection first, focus on protecting and rehabilitating the lake ecosystem, rationally develop the area around the lake, strictly implement the scientific plan, and finally ensure that the towns around the Poyang Lake eco-economic zone have Coordinated development. The comprehensive development and governance of the Great Lakes Basin is a major issue in the world today. The research work of this paper can provide some reference for the content that needs to be analyzed in the comprehensive development and governance of the Great Lakes Basin, that is, the characteristics of urban land expansion and evolution in the region.